# Optimization of Transverse OAM Transmission through Few-Mode Fiber

Chong Zhang [1], Qian Cao [1,2,*] and Qiwen Zhan [1,2]

1   School of Optical-Electrical and Computer Engineering, University of Shanghai for Science and Technology, Shanghai 200093, China; 213330644@st.usst.edu.cn (C.Z.); qwzhan@usst.edu.cn (Q.Z.)
2   Zhangjiang Laboratory, 100 Haike Road, Shanghai 201204, China
*   Correspondence: cao.qian@usst.edu.cn

**Abstract:** Spatiotemporal optical vortex (STOV) wavepacket is a new type of vortex optical field carrying transverse orbital angular momentum (OAM). Due to the presence of imbalanced dispersion and diffraction phase, the STOV pulse undergoes fragmentation during free space propagation, leading to the disappearance of the spatiotemporal vortex phase structure. For practical applications, having a stable long-distance propagation of STOV pulse is critical. Recent work demonstrates the transmission of transverse OAM in few-mode fiber. However, the maximum transmission distance is limited to 100 cm due to excessive group velocity dispersion between modes. In this work, we optimize the transmission of transverse OAM by engineering fiber parameters. By tuning the radius of the fiber core and the relative refractive index difference, the group time delay difference values between the $LP_{01}$ and $LP_{11}$ modes and their corresponding group velocity dispersion coefficients are minimized. The simulation results show that the optimized fiber allows the first-order STOV pulse to propagate up to 500 cm, and the second-order STOV pulse up to 300 cm without distorting the spatiotemporal vortex phase structure. Long-distance propagation of STOV pulse can create new opportunities and facilitate applications such as developing novel transverse OAM lasers and telecommunication approaches.

**Keywords:** spatiotemporal optical vortex beam; transverse orbital angular momentum; spatiotemporal vortex phase

## 1. Introduction

Vortices are widespread in nature phenomena, appearing as swirling clouds in the atmosphere, tornadoes formed by the high-speed rotation of cyclonic structures, and swirling currents in the ocean. In recent decades, optical vortices have received significant attention in the field of optics. Vortex beams, characterized by a helical phase structure in the spatial domain, have become a classic tool for studying these properties due to their unique characteristics and ease of generation. The beam can possess spin angular momentum (SAM) and orbital angular momentum (OAM) [1]. In 1992, Allen et al. discovered that OAM is an inherent property of vortex light with a spiral phase-front, described by a phase structure of $e^{il\theta}$, carrying an orbital angular momentum equivalent to $l\hbar$ per photon (where $l$ is an integer number) [2]. OAM beams have been widely utilized in quantum optics [3–5], nano-optics [6–8], biomedicine [9,10], optical communication [11–14], and microscopic imaging [15–17].

Among the most common types of optical vortices, optical vortex beams carrying longitudinal OAM are already well known, with the OAM orientation being parallel or antiparallel to the direction of light propagation. In contrast, recent theoretical studies have indicated that a spatiotemporal optical vortex (STOV) wavepacket can carry transverse OAM with its OAM orientation perpendicular to the direction of propagation. Recently, researchers have successfully generated a spatiotemporal optical vortex with controllable

purely transverse OAM by using a high-resolution spatial light modulator (SLM) placed in the spatial–spectral domain of the light field [18]. In the space–time domain ($x - t$ plane), STOV features a vortex phase structure. The spatiotemporal coupling property of STOV confers it with a complex optical field morphology, thereby opening up a new research avenue for manipulating pulsed optical beams and exploring nonlinear fiber optical phenomena that involve OAM. The propagation of STOV in free space is influenced by diffraction, which leads to the distortion of the donut-shaped intensity profile in the spatiotemporal domain. This distortion is primarily attributed to diffraction-induced phase effects [19,20]. Similarly, when STOV is transmitted in dispersive media, the presence of temporal dispersion poses a challenge in maintaining the vortex phase structure of the spatiotemporal domain over long transmission distances [21]. The abovementioned effects impose constraints on the long-distance transmission of STOV pulses, and limit the scope for their potential application development.

To overcome the challenges associated with long-distance STOV pulse propagation and the preservation of the spatiotemporal vortex phase characteristics exhibited by STOV pulses, the utilization of step-index, few-mode fiber (FMF), namely, SMF-28, as the transmission medium is proposed and experimentally demonstrated at the wavelength of 1030 nm [22]. Under the weakly guidance assumption, the STOV pulse coupled into a fiber can be decomposed into two LP modes, namely, the $LP_{01}$ and $LP_{11}$ modes. However, the propagation distance of a STOV pulse inside the FMF is still limited to about one meter due to the group velocity mismatch between LP modes. In this study, we perform numerical simulation of STOV pulse propagation inside commercially available SMF-28 fiber. We propose an optimization strategy of the fiber by engineering the key parameters of SMF-28 fiber so that the STOV pulse can extend the propagation distance by a factor of 5. Compared with conventional SMF-28 fiber, this optimized optical fiber effectively mitigates the group velocity mismatch between the LP modes, guaranteeing the preservation of the spatiotemporal vortex phase attributes of STOV pulses over extended transmission distances.

## 2. Materials and Methods

In the spatiotemporal domain, a spatiotemporal optical vortex (STOV) pulse with topological charge $l$ typically exhibits a donut-shaped intensity distribution. This pattern features the highest intensity in a ring-shaped region surrounding the vortex core, with the intensity gradually decreasing towards the central and outer regions. Simultaneously, the beam's phase depicts a helical motif, revolving around the central vortex core. The STOV pulse with $l = 2$ exhibits two vortices in the phase distribution in the spatiotemporal domain. Under the weak waveguide approximation, the first-order and second-order STOV pulses with a center wavelength of 1030 nm can be decomposed into $LP_{01}$ and $LP_{11}$ modes after coupling into the fiber. The STOV pulse is transmitted through the combined superposition of $LP_{01}$ and $LP_{11}$ modes, as expressed in the following equation:

$$E_{\text{STOV}(l=1)}(x,t) = \widetilde{c}_1(t)E_{01}(x) + \widetilde{c}_2(t)E_{11}(x) \tag{1}$$

$$E_{\text{STOV}(l=2)}(x,t) = \widetilde{c}_3(t)E_{01}(x) + \widetilde{c}_4(t)E_{11}(x) \tag{2}$$

$E_{01}(x)$ and $E_{11}(x)$ denote the normalized electric field of the $LP_{01}$ and $LP_{11}$ modes in the optical fiber. $\widetilde{c}_1$ and $\widetilde{c}_2$ describe the coefficients of $LP_{01}$ and $LP_{11}$ modes in time. These two variables vary in time and evolve during the pulse propagation. $t$ represents the localized time elapsed after the STOV pulse is coupled into the optical fiber. The complex numbers $\widetilde{c}_1$ and $\widetilde{c}_2$ are obtained in Equations (3) and (4) by integrating the LP modes eigenfunction with the STOV pulse over the variable $x$.

$$\widetilde{c}_1(t) = \frac{\int E_{\text{STOV}(l=1)}(x,t)E_{01}^*(x)dx}{\int E_{01}(x)E_{01}^*(x)dx} \tag{3}$$

$$\widetilde{c}_2(t) = \frac{\int E_{\text{STOV}(l=1)}(x,t)E_{11}^*(x)dx}{\int E_{11}(x)E_{11}^*(x)dx} \tag{4}$$

Figure 1a shows the intensity and phase distribution of a spatiotemporal vortex beam with $l = 1$, a pulse width of 160 fs and a beam size of 5.7 µm synthesized by the $LP_{01}$ and $LP_{11}$ modes in the spatiotemporal domain, where a vortex phase exists. Figure 1(b1,b2) illustrates the real and imaginary parts of $\widetilde{c}_1$ and $\widetilde{c}_2$ over time. The same treatments are carried out for solving the values of $\widetilde{c}_3$ and $\widetilde{c}_4$ in the synthesized second-order spatiotemporal vortex pulse. Figure 1(b3,b4) shows the real and imaginary parts of $\widetilde{c}_3$ and $\widetilde{c}_4$. Figure 1c shows a spatiotemporal vortex beam with $l = 2$ synthesized by the $LP_{01}$ and $LP_{11}$ modes. It exhibits two vortex phases in the spatiotemporal domain with opposite rotational directions.

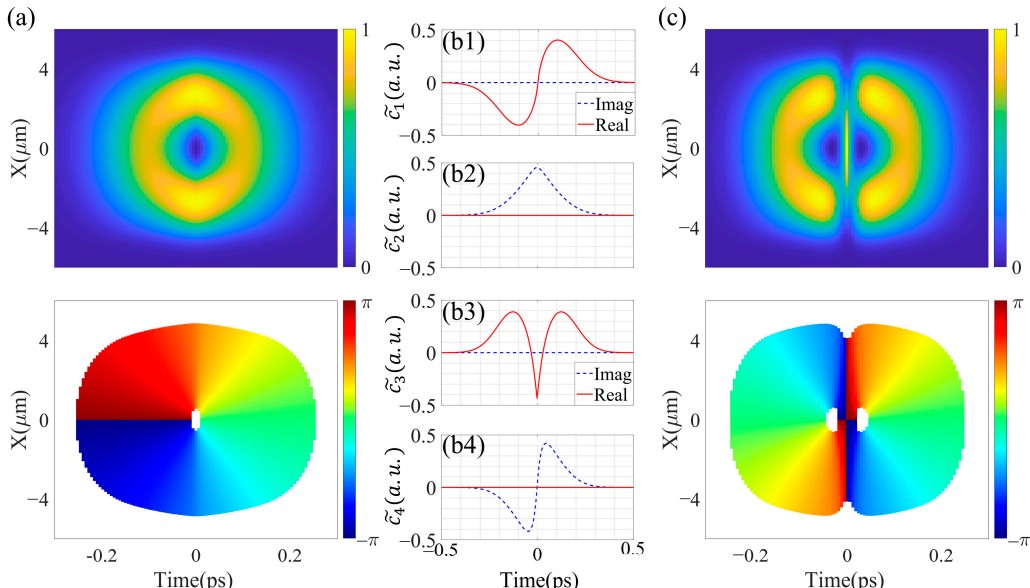

**Figure 1.** Modal decomposition of STOVs in LP modes. (**a**) The intensity and phase of the STOV ($l = 1$) synthesized by the $LP_{01}$ and $LP_{11}$ modes. (**b1,b2**) Complex coefficients for $LP_{01}$ mode and $LP_{11}$ mode for synthesizing a STOV pulse with $l = 1$. (**b3,b4**) Complex coefficients of $LP_{01}$ and $LP_{11}$ modes for synthesizing a STOV ($l = 2$). (**c**) The intensity and phase of a STOV ($l = 2$) synthesized from the $LP_{01}$ and $LP_{11}$ modes.

As the STOV pulse is synthesized from the $LP_{01}$ and $LP_{11}$ modes, such treatment requires a proper selection of an optical fiber that can simultaneously support both the $LP_{01}$ and $LP_{11}$ modes. For fiber, it is necessary to have a V-value that is greater than 2.405. The normalized frequency value of the optical fiber is expressed as $V = k_0 a n_1 \sqrt{2\Delta}$, where $a$ is the radius of the optical fiber core, $n_1$ is the refractive index of the core, and $n_2$ is the refractive index of the cladding. The relative refractive index difference $\Delta$ expresses the relationship between $n_1$ and $n_2$:

$$\Delta = \frac{n_1^2 - n_2^2}{2n_1^2} \tag{5}$$

## 3. Results and Discussion

Figure 2a depicts the cross-sectional structure and refractive index distribution of the step-index fiber. SMF-28 optical fiber is taken as the basis for optimization and analysis, with a core radius of 4.1 µm and a relative refractive index difference of 0.36% between the core and cladding. The STOV pulse is transmitted as a combination of the $LP_{01}$ and $LP_{11}$ modes in the optical fiber. The propagation constants $\beta_{01}$ for the $LP_{01}$ mode and $\beta_{11}$ for the $LP_{11}$ mode can be obtained by solving their individual normalized eigenfunction. The group delay value can then be solved as $\beta_1 = \partial\beta/\partial\omega = (v_g)^{-1}$, and the group velocity

dispersion coefficient (GVD) denoted as $\beta_2 = \partial^2\beta/\partial\omega^2$. The two modes propagate in the optical fiber with different group velocities, $v_g$, respectively. $\Delta T$ represents the group delay difference between the $LP_{01}$ mode and $LP_{11}$ modes for 1 m propagation inside the fiber, where $\Delta T = |\beta_{1-LP11} - \beta_{1-LP01}|$ [23]. As the STOV pulse propagates through the fiber over a greater distance, the group delay difference between the $LP_{01}$ and $LP_{11}$ modes gradually increases. GVM can cause a gradual separation of the two modes as they propagate. Excessive values of $\Delta T$ and GVD can both impact the propagation of STOV, leading to the breakup of its vortex phase in the spatiotemporal domain.

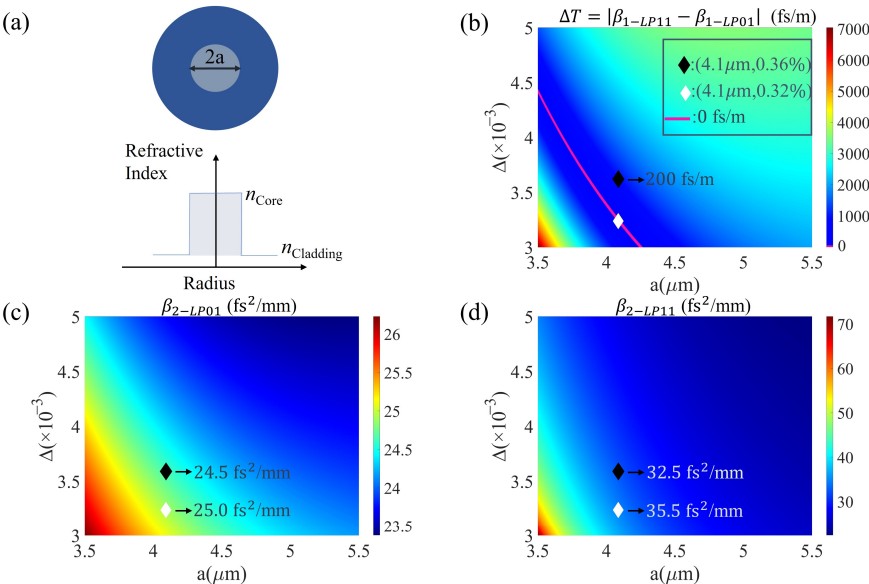

**Figure 2.** The cross-sectional structure and refractive index distribution of the step-index fiber. (**a**) The cross-sectional structure of the step-index fiber. (**b**) Absolute value distribution of group delay differences between $LP_{01}$ mode and $LP_{11}$ mode. (**c**) Distribution of GVD coefficient values for $LP_{01}$ mode. (**d**) Distribution of GVD coefficient values for $LP_{11}$ mode.

In order to preserve the integrity of the vortex phase, it is therefore necessary that the group delay difference $\Delta T$ and GVD are optimized to their minimum. We calculate the fiber by substituting the core radius in the range of 3.5 to 5.5 μm and the relative refractive index difference between the fiber core and cladding in the range of 0.3% to 0.5%. From the normalized eigenfunction of both $LP_{01}$ and $LP_{11}$ mode, one can subsequently calculate the group delay differences for different combinations of core radius and relative refractive index difference, along with the corresponding GVD coefficients for the $LP_{01}$ and $LP_{11}$ modes.

Figure 2b illustrates the distribution of $\Delta T$ within the core radius range of 3.5 to 5.5 μm and relative refractive index difference ranging from 0.3% to 0.5% between the fiber core and cladding. Figure 2c illustrates the GVD coefficient values corresponding to the $LP_{01}$ mode for different combinations of core radius and relative refractive index difference. Similarly, Figure 2d displays the GVD coefficient values corresponding to the $LP_{11}$ mode for various combinations of core radius and relative refractive index difference.

In Figure 2b, the highlighted red region represents a set of optimal core radius and relative refractive index difference combinations of core radius and relative refractive index difference. This region serves as a significant reference for systematically devising rational strategies in designing the core radius and relative refractive index difference of optical fibers. A well-considered configuration of these parameters can lead to the achievement of a minimized group delay difference $\Delta T$ between the $LP_{01}$ and $LP_{11}$ modes, approaching nearly 0 fs/m. From the graph, while maintaining a constant core radius, the group delay difference $\Delta T$ increases with an increasing relative refractive index difference. For the SMF-28 optical fiber with the established parameters (core radius of 4.1 μm, relative

refractive index difference of 0.36%), the corresponding $\Delta T$ value is about 200 fs/m. To mitigate the group delay difference, an optimized fiber design is achieved by maintaining the core radius at 4.1 μm while setting the relative refractive index difference to 0.32%. Consequently, the optimized fiber configuration results in a reduced $\Delta T$ value of 0 fs/m.

In Figure 2c, when the core radius is set at 4.1 μm, the GVD coefficient values for the $LP_{01}$ mode decrease as the relative refractive index increases. Similarly, in Figure 2d, the GVD coefficient values for the $LP_{11}$ mode also decrease as the relative refractive index increases. Minimizing group velocity mismatch (GVM) and GVD is essential to ensure that STOV pulses maintain their vortex phase even after undergoing long-distance propagation in optical fibers. The optimal fiber design involves achieving simultaneously small values for both $\Delta T$ and GVD coefficient. The significant disparity in group delay between the $LP_{01}$ and $LP_{11}$ modes has a more noticeable effect on preserving the vortex phase in the spatiotemporal domain after propagation compared to the influence of the GVD coefficients. While the optimized fiber exhibits a marginal increase in the GVD coefficient values for the $LP_{01}$ and $LP_{11}$ modes, it significantly reduces the group delay difference between the two modes in comparison to SMF-28.

By coupling the first-order and second-order STOV pulses into both SMF-28 optical fiber and the optimized fiber, we simulate the propagation of the STOV pulses for a distance of 100 cm, 300 cm, and 500 cm. A comparative analysis was performed on the intensity and phase distributions of the STOV pulses in the spatiotemporal domain after propagating through two types of fibers. Assuming a central wavelength of 1030 nm for the STOV pulse coupled into the fiber for propagation, and considering linear propagation without any loss within the fiber, the scenario assumes that there is no cross-talk between the $LP_{01}$ and $LP_{11}$ modes. In Figure 2b–d, the black diamond sign represents SMF-28 fiber and the white diamond sign represents the optimized optical fiber. From Figure 2b–d, the group delay difference $\Delta T$ between the $LP_{01}$ mode and $LP_{11}$ mode in SMF-28 is 200 fs/m, the GVD of the $LP_{01}$ mode is 24.5 fs$^2$/mm, and the GVD of the $LP_{11}$ mode is 32.5 fs$^2$/mm. Figure 3a shows the propagation results of the first-order STOV pulse in SMF-28 and $z$ represents the length of propagation within the optical fiber; as the propagation distance increases, the simulated intensity in the spatiotemporal domains gradually broadens and disperses, losing its original profile. Upon propagating for a distance of 300 cm, the vortex phase remains preserved. However, due to the notable group delay difference between the $LP_{01}$ and $LP_{11}$ modes, the vortex phase undergoes a breakup after propagating for 500 cm. Figure 3b depicts the propagation results of a second-order STOV pulse in SMF-28. With increasing the propagation distance, the previously symmetric intensity distribution in the spatiotemporal domain gradually becomes asymmetrical. After propagating for 100 cm, the phase of the two counter-rotating vortices remains preserved; however, the vortex phase undergoes a breakup after propagating for 3 m.

In comparison, the group delay difference $\Delta T$ between the $LP_{01}$ mode and $LP_{11}$ mode in the optimized fiber (4.1, 0.32%) is reduced to nearly 0 fs/m. The GVD of $LP_{01}$ mode is 25.0 fs$^2$/mm, and the GVD of $LP_{11}$ mode is 35.5 fs$^2$/mm. Figure 3c presents the outcomes of the first-order STOV pulse propagation at various distances within the optimized fiber; the intensity distribution in the spatiotemporal domain broadens during the propagation process. Compared to SMF-28 fiber, the spatiotemporal vortex phase of the STOV pulse remains stable in the optimized fiber even after propagating for 500 cm. The propagation distance has been significantly extended. Figure 3d shows the propagation results of the second-order STOV pulse in the optimized optical fiber. The intensity remains symmetrical after propagating for a distance of 300 cm, and two vortex phases with opposite rotational directions are still retained in the spatiotemporal domain. The vortex phase breaks up after propagating for 500 cm. First-order and second-order STOV pulses maintain the main characteristics of their vortex phases in the spatiotemporal domain when transmitted through the optimized optical fiber for distances exceeding 300 cm.

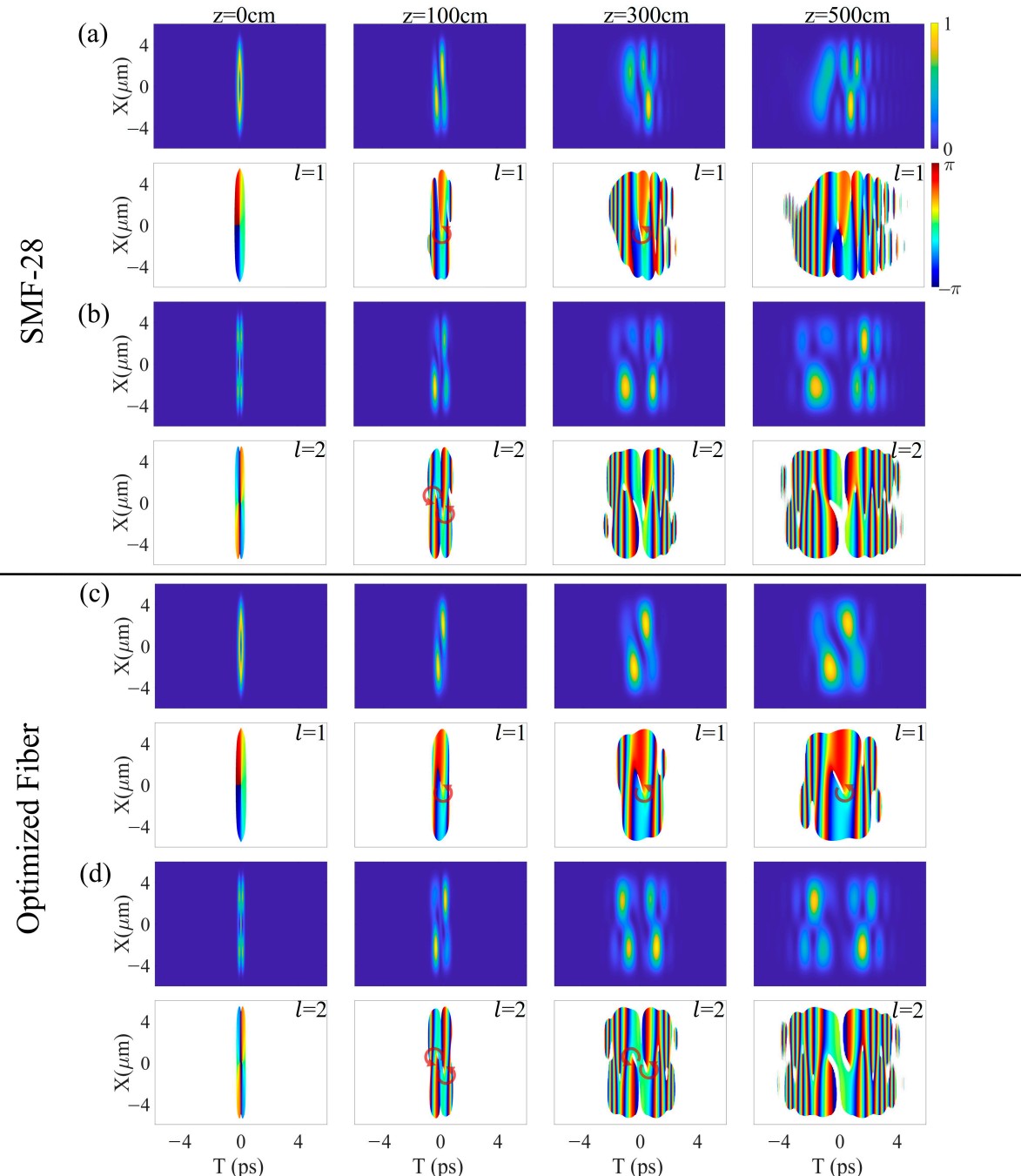

**Figure 3.** Numerical propagation of synthesized STOV pulse ($l = 1$, $l = 2$) in SMF-28 fiber ($a = 4.1$ μm, $\Delta = 0.36\%$) and optimized fiber ($a = 4.1$ μm, $\Delta = 0.32\%$). (**a**) Synthesized STOV pulse ($l = 1$) in SMF-28 fiber. (**b**) Synthesized STOV pulse ($l = 2$) in SMF-28 fiber. (**c**) Synthesized STOV pulse ($l = 1$) in optimized fiber. (**d**) Synthesized STOV pulse ($l = 2$) in optimized fiber.

## 4. Conclusions

In conclusion, we performed a numerical study on the optimization of transverse OAM inside few-mode fiber. By engineering fiber parameters, the maximum transmission length of transverse OAM can be extended to 500 cm, which is five times larger than the previous reported results. The group delay difference between the $LP_{01}$ mode and the $LP_{11}$ mode of this optimized fiber can be minimized. Such fiber can significantly enhance the stability of the STOV pulse and prevent breaking-up in pulse propagation. This study paves the way for a new method for long-distance propagation of STOV pulses and can

facilitate many potential applications using STOV pulses such as novel telecommunications and building transverse OAM lasers.

**Author Contributions:** Methodology, Q.C.; investigation, C.Z. and Q.C; writing—original draft preparation, C.Z.; writing—review and editing, Q.C. and Q.Z.; supervision, Q.C. and Q.Z. All authors have read and agreed to the published version of the manuscript.

**Funding:** We acknowledge the support from the National Natural Science Foundation of China (NSFC) [grant Nos. 92050202 (Q.Z.) and 12104309 (Q.C.)], the Shanghai Science and Technology Committee [grant No. 19060502500 (Q.Z.)], and the Shanghai Sailing Program [grant No. 21YF1431500 (Q.C.)].

**Institutional Review Board Statement:** Not applicable.

**Informed Consent Statement:** Not applicable.

**Data Availability Statement:** The data and code for the numerical calculations that support the findings of this study are available from the corresponding author upon reasonable request.

**Conflicts of Interest:** The authors have no conflicts of interest to disclose.

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
