# Peer review of "Optimization of Transverse OAM Transmission through Few-Mode Fiber"

_photonics, doi:10.3390/photonics11040328_

Round 1

Reviewer 1 Report

Comments and Suggestions for Authors

In this paper, the authors have optimized the SMF-28 fiber and performed simulations for the transmission of spatiotemporal vortex beams in an optimized fiber. The authors have novelty extended the effective propagation distance of STOV and simulated second-order STOV propagation in optical fibers. This work brings a fresh and novel perspective to the applications of spatiotemporal optical vortices, the manuscript is worthy of publication for Photonics. However, I do have the following brief comments to revise before publication.

 1. The authors mention the GVM effect affecting the propagation of STOV pulses in optical fibers, how does it affect propagation? Please further clarify this point.

2. According to Eq.(1) and Eq.(2) , are the parameters of the first-order and second-order STOV pulses coupled into the fiber the same? Such as, their pulse width and spatial dimensions?

3. Please further explain the main differences between STOV pulse propagation in optical fibers and in free space.

Author Response

Dear reviewer of our manuscript,

First of all, we would like to express our gratitude to you for your time spent on reviewing the manuscript. Based on the review reports, we revised the manuscript carefully and sincerely hope the revision has satisfyingly addressed the issues raised by the reviewers. Attached below is our point-by-point responses to the comments of the reviewers.

In this paper, the authors have optimized the SMF-28 fiber and performed simulations for the transmission of spatiotemporal vortex beams in an optimized fiber. The authors have novelty extended the effective propagation distance of STOV and simulated second-order STOV propagation in optical fibers. This work brings a fresh and novel perspective to the applications of spatiotemporal optical vortices, the manuscript is worthy of publication for Photonics. However, I do have the following brief comments to revise before publication.

Reply: We thank the reviewer for his/her positive response on our manuscript.

Comment 1: The authors mention the GVM effect affecting the propagation of STOV pulses in optical fibers, how does it affect propagation? Please further clarify this point.

Reply: The STOV pulse travels along the fiber as a blend of LP01 and LP11 modes. GVM characterizes the phenomenon where the group velocities of these two LP modes are different. This can result in a gradual separation of the modes during propagation. Excessive GVM can thus distort the STOV pulse and eventually cause its disintegration. This explanation has been added in the first paragraph in Section 3 in the revised manuscript.

Comment 2: According to Eq.(1) and Eq.(2) , are the parameters of the first-order and second-order STOV pulses coupled into the fiber the same? Such as, their pulse width and spatial dimensions?

Reply: In the simulation, the parameters of the first and second order STOV are the same except for the topological charge for the spatiotemporal vortex phase. The input pulse has a pulse duration of about 160 fs and a beam size of around 5.7 μm. This has been added in the third paragraph in Section 2 in the revised manuscript.

Comment 3: Please further explain the main differences between STOV pulse propagation in optical fibers and in free space.

Reply: When the STOV pulse propagates in free space, the STOV pulse gradually splits in spatial direction because of diffraction. The pulse can become severely distorted and lose its integrity after a long propagation distance. In contrast, when STOV pulses are propagating inside an optical fiber, the diffraction effect is absent because of the waveguiding effect. The STOV pulse, therefore, is tightly confined within the fiber and can propagate for a long distance without disintegrating itself.

Sincerely Yours,

Qian Cao on behalf of all authors.

Reviewer 2 Report

Comments and Suggestions for Authors

This manuscript aims to optimize the propagation length of a spatiotemporal optical vortex beam in a fiber by tuning the geometry of the fiber. The study is solid and the results are well explained. I do not have technical comments regarding the content. But just a general concern regarding the manuscript, the authors may want to emphasize their novelty and motivation on making this optimization. 

By aiming at the geometry, such as the core size of shell size of the fiber, how easy can people to obtain this optimized size of fiber, and how precise the parameters need to be to make the numerical simulation results stay valid?

Also, how robust their simulation is against fiber bending?

Author Response

Dear reviewer,

First of all, we would like to express our gratitude to you for your time spent on reviewing the manuscript. Based on the review reports, we revised the manuscript carefully and sincerely hope the revision has satisfyingly addressed the issues raised by the reviewers. Attached below is our point-by-point responses to the comments of the reviewers.

This manuscript aims to optimize the propagation length of a spatiotemporal optical vortex beam in a fiber by tuning the geometry of the fiber. The study is solid and the results are well explained. I do not have technical comments regarding the content. But just a general concern regarding the manuscript, the authors may want to emphasize their novelty and motivation on making this optimization.

Reply: We thank the reviewer for his/her positive response on our manuscript.

Comment 1: By aiming at the geometry, such as the core size of shell size of the fiber, how easy can people to obtain this optimized size of fiber, and how precise the parameters need to be to make the numerical simulation results stay valid?

Reply: In fiber drawing process, the uncertainty of the fiber geometry can be well controlled within less than 1%. Our study has suggested an optimization for STOV pulse propagation by engineering the refractive index difference () of the fiber. This can be achieved by controlling the doping concentration when the fiber preform is manufactured. As long as the uncertainty of is controlled within 3% ( for ), the group velocity mismatch offered by the optimized fiber can extend the maximum propagation distance of STOV pulse.

Comment 2: Also, how robust their simulation is against fiber bending?

Reply: The simulation approach is robust as long as the bending of the fiber is not severe. In our previous work, a 1-meter long was bent in a radius of around 20 cm for a total amount of 90 degree angle without deteriorating the transmission of STOV pulse. In this study, the mode field radius of LP01 mode for our optimized fiber (3.94 μm) is slightly larger (about 3%) than that of an un-optimized fiber (3.82 μm). The mode field radius is still significantly smaller than the radius of the fiber cladding (normally, it is 67.5μm). We believe this large contrast between the mode size and the fiber size can guarantee the validity of our simulation.

Sincerely Yours,

Qian Cao on behalf of all authors.